Review

Subject Area:
genetics/genomics/molecular biology

Keywords:
transposon regulation, piRNA, adaptive evolution, host–pathogen arms race, pathogen mimicry

Author for correspondence:
William E. Theurkauf
e-mail: william.theurkauf@umassmed.edu

# Rapid evolution and conserved function of the piRNA pathway

Swapnil S. Parhad and William E. Theurkauf

Program in Molecular Medicine, University of Massachusetts Medical School, 373 Plantation Street, Worcester, MA 01605, USA

 SSP, 0000-0002-5723-9384; WET, 0000-0001-7342-1912

Transposons are major genome constituents that can mobilize and trigger mutations, DNA breaks and chromosome rearrangements. Transposon silencing is particularly important in the germline, which is dedicated to transmission of the inherited genome. Piwi-interacting RNAs (piRNAs) guide a host defence system that transcriptionally and post-transcriptionally silences transposons during germline development. While germline control of transposons by the piRNA pathway is conserved, many piRNA pathway genes are evolving rapidly under positive selection, and the piRNA biogenesis machinery shows remarkable phylogenetic diversity. Conservation of core function combined with rapid gene evolution is characteristic of a host–pathogen arms race, suggesting that transposons and the piRNA pathway are engaged in an evolutionary tug of war that is driving divergence of the biogenesis machinery. Recent studies suggest that this process may produce biochemical incompatibilities that contribute to reproductive isolation and species divergence.

## 1. Introduction

Single celled organisms to complex animals face the threat of pathogens, which are countered by powerful adaptive and innate immune systems [1]. However, the targets of host defence systems can mutate to evade detection or express inhibitors that suppress the host immune response [2]. Host–pathogen interactions thus lead to the positive selection of pathogen mutations that evade the host defences and allow propagation, followed by positive selection of host mutations that restore the pathogen control. The resulting 'Red Queen arms race', characterized by cycles of adaptive evolution, drives rapid coevolution of interacting host and pathogen genes [3]. Transposons are integral genome constituents that can mobilize and cause mutations and genomic instability, and the Piwi-interacting RNA (piRNA) pathway functions as the host defence against these pathogens [4,5]. Many piRNA pathway genes show evidence of adaptive evolution [6], suggesting that they are engaged in an arms race with the transposons they control. Here we contrast the conserved mechanisms that drive transposon replication with the divergent processes that produce the piRNAs that silence them and speculate that this is the product of a never-ending arms race that may have profound evolutionary consequences.

### 1.1. Diverse transposons, conserved transposition mechanisms

Transposons were discovered by Barbara McClintock through cytogenetic analysis of mosaic pigmentation patterns in maize kernels [7,8]. Since this initial finding, transposons have been found in essentially every organism [9,10]. They are also remarkably diverse. For example, there are over 120 transposon families in *Drosophila melanogaster*. However, these diverse mobile elements move by a limited number of transposition mechanisms, which use either DNA or RNA intermediates (figure 1) [11,12].

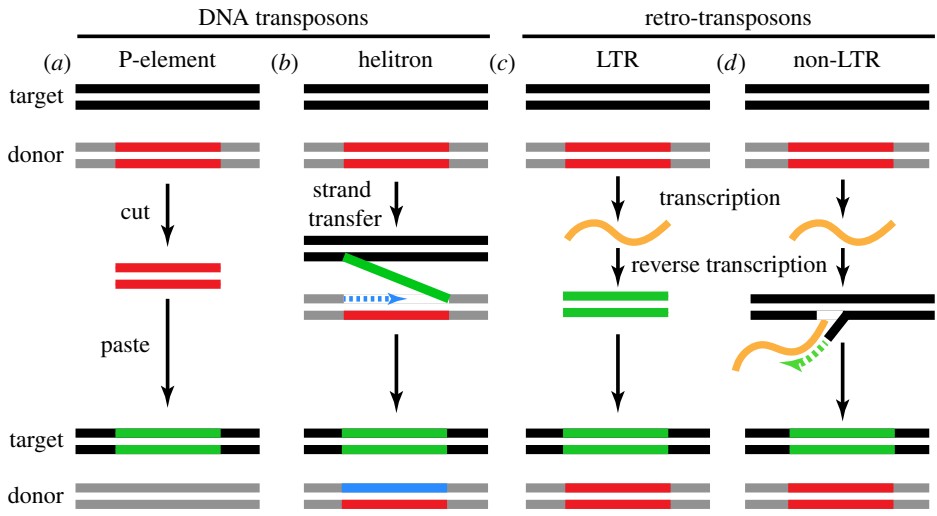

**Figure 1.** Summary of transposition mechanisms. Transposition mechanisms for major eukaryotic transposons. (*a,b*) DNA transposons which transpose through a DNA intermediate. (*c,d*) RNA or retrotransposons which transpose through an RNA intermediate. Target and donor sites are shown in black and grey, respectively. Old and new transposable elements (TEs) are shown in red and green, respectively. Examples of such transposons are denoted. (*a*) DNA transposons, such as P-elements, excise from the donor and insert into a new site. (*b*) Helitrons transfer one DNA strand from the donor to the recipient site. The donor site synthesizes a new strand (shown in blue). The recipient site also synthesizes a new strand. (*c*) LTR retrotransposons transcribe into an RNA. This RNA is reverse-transcribed and inserted into a new site. (*d*) Non-LTR retrotransposons also transcribe into an RNA. The RNA is reverse-transcribed at the insertion site. Thus, the original donor site is unaffected for retrotransposons.

DNA transposable elements move by a cut–paste mechanism and encode a transposase that recognizes inverted terminal repeats and catalyses excision of an existing element and integration of the excised double stranded DNA intermediate into a new site [11]. Examples of such DNA transposons are P-elements in *Drosophila* and Tc1 elements in *Caenorhabditis elegans* [13,14]. Helitrons, another type of DNA transposon, move through a single-stranded DNA intermediate and leave the donor element intact [15,16]. The helitron transposase nicks one end of a donor element and the target site. The 3′-end of the target nick is ligated to the 5′-end of the donor element, and replication from the 3′-end of the donor nick displaces one strand of the transposon and generates a new second strand. A second nick releases the 3′-end of the displaced strand, which is ligated to the 5′-end of the target site, forming an acceptor site heteroduplex with a loop containing the new helitron (not shown). Replication of this chromosome generates one strand carrying the acceptor site and one strand with a new copy of the element (figure 1).

Retrotransposons move by a copy–paste mechanism with an RNA intermediate [11]. These elements are related to retroviruses and encode a reverse transcriptase that makes a DNA copy from a transposon transcript, which is integrated into a new site. Retrotransposons are further subdivided by structure and replication capacity. Elements that have long terminal repeats and encode reverse transcriptase are termed LTR retrotransposons and include Ty elements in *Saccharomyces cerevisiae* and Burdock from *Drosophila* [17,18]. A subset of these retrotransposons encodes gag, pol and env proteins and can make viral particles [19]. These endogenous retroviruses can move from cell to cell, or from animal to animal, leading to horizontal transfer. For example, *Drosophila gypsy* and *ZAM* elements expressed in the somatic follicle cells of the ovary can infect adjacent germline cells [20,21]. Another subset of retrotransposons lacks LTRs and is classified as long interspersed elements (LINEs) or short interspersed elements (SINEs). LINEs are autonomous and encode a reverse transcriptase and endonuclease that

mediates transposition, while SINEs are non-autonomous and use LINE encoded enzymes to move. Jockey from *D. melanogaster* and L1 in mammals are examples of non-LTR retrotransposons [22,23]. From single-celled organisms to complex animals, transposons move by this limited set of mechanisms, mediated by enzymes with conserved biochemical functions.

## 1.2. Transposons as pathogens

Transposons can disrupt the host genome function by a variety of mechanisms. Transposition into exons disrupts the coding sequence, and intron insertions can alter the splicing patterns and generate novel and potentially deleterious fusion proteins [5]. Promoter or enhancer insertions can change the gene transcription, whereas insertions in 5′- or 3′-UTRs can affect the post-transcriptional gene regulation. Transposition also leads to DNA nicks and double-strand breaks, and errors in the repair of these lesions can lead to recombination between transposon repeats, triggering chromosomal duplications, deletions, translocations and inversions [24]. Consistent with these observations, transposition has been linked to cancer and many other diseases [5,25]. Intriguingly, activation of the *Steamer* retrotransposon has been linked to clonal cancer cells that are transmitted between clams, leading to horizontal spread through wild populations [26]. Limiting transposition is therefore essential to maintaining normal cell function.

Most transposons cannot exit the cell and are propagated through replication in germ cells, which leads to vertical transmission of new genomic copies. By contrast, endogenous retroviruses, described above, can assemble virus-like particles and infect new hosts. However, horizontal transfer of DNA transposable elements (TEs), which do not form infectious particles, has been observed [27,28]. These events can occur between distantly related species. For example, P-elements are DNA transposable elements that recently moved from *Drosophila willistoni* into *D. melanogaster*. These species are separated by approximately 50 Myr, but the

P-elements they harbour differ only by one nucleotide [29]. By contrast, Piwi, which binds the piRNAs that silence P-elements, shows 33% amino acid sequence divergence between these species. The mechanisms leading to horizontal transfer of DNA elements are not understood, but DNA transposons and retrotransposons in distinct *Drosophila* species generally show less sequence divergence than protein coding genes [30–32], which appears to reflect horizontal transfer. Similar patterns are observed in other animals and plants (reviewed in [28]). For example, SPIN family transposons appear to have undergone horizontal transfer between mammals and tetrapods [33], and Tc1 like transposons have moved between fish and frogs [34]. Horizontal TE transfer thus appears to be widespread, reflected in the conservation of mobile elements between species with significant protein coding gene divergence.

# 2. The piRNA host immune defence against transposons

Animals produce small piRNAs to control transposons during the germline development [35]. With exogenous viruses or bacteria, the immune response is mounted after infection. The piRNA pathway, by contrast, must continuously suppress TEs that are integral genome components and respond to the invasion of new elements. piRNA biogenesis and function have been extensively studied in flies [36], mice [37] and worms [38], and key components have been characterized in planarians [39], fish [40], chickens [41] and humans [42]. Functional studies in flies, worms and mice define a conserved role for piRNAs in transposon control, but also highlight the diversity of the silencing machinery.

piRNAs were identified in *Drosophila*, through an analysis of *Stellate* (*Ste*) silencing by the *Suppressor of Stellate* (*Su(Ste)*) locus [43]. In this system, the mutation in *Su(Ste)* leads to male sterility and over-expression of Ste protein, which assembles into prominent needle-like crystals in the testes [44,45]. Aravin *et al.* [43] showed that *Su(Ste)* encodes short RNAs that are complementary to *Ste,* and that mutations in *SpnE,* subsequently shown to be required for piRNA production, lead to over-expression of Ste and a subset of transposons. Subsequent analysis of the tissue distribution of short RNAs, performed by direct cloning and sequencing, identified 23–30 nt long RNAs matching transposons in germline tissue. While miRNAs and siRNAs are produced from double stranded precursors by Dicer endonuclease cleavage, production of these germline enriched small RNAs is Dicer independent [35]. Similar small RNAs were subsequently found in mouse testes [46,47] and shown to bind to the mouse homologues of Piwi, a *Drosophila* gene required for germline development [48,49]. Piwi is the founding member of the PIWI clade of Argonaute proteins, and these novel small RNAs were therefore named Piwi-interacting RNAs (piRNAs).

## 2.1. piRNA biogenesis in *Drosophila*

In flies, mutations that disrupt the piRNA transposon silencing system lead to female sterility and defects in embryonic patterning which can be easily quantified by visual inspection of the eggs produced by mutant females [50]. At the time when piRNAs were first described, maternal genetic control of embryonic patterning was a mature field, but the molecular functions of many of the genes required for embryonic patterning were not well understood [51–53]. A major breakthrough came with the realization that germline genome instability, and activation of damage signalling through ATR and Chk2 kinases, trigger embryonic patterning defects [54–56]. These initial findings were based on an analysis of meiotic repair mutants, but the subsequent study showed that the patterning defects in several piRNA pathway genes are also caused by Chk2 activation [50]. These findings argued that transposon silencing is likely the primary function for *Drosophila* piRNAs and suggested that previously identified patterning mutations would identify new piRNA pathway genes [51,52]. Established genetic resources, with new genome-wide screens for mutations triggering patterning defects and transposon over-expression [57–59], thus led to the rapid identification of the machinery that produces piRNA precursors, processes these long RNAs into mature piRNAs and silences their targets.

### 2.1.1. Primary piRNA biogenesis

The primary piRNAs that initiate transposon silencing are derived from specific genomic loci, called piRNA clusters, composed of nested transposon insertions, which function as an archive of transposon sequences (figure 2*a*) [60,61]. *Drosophila* ovaries are composed of cysts containing 15 germline nurse cells and one oocyte, surrounded by a monolayer of somatic follicle cells. In the germline, the predominant clusters contain randomly oriented transposon arrays and produce piRNAs from both genomic strands. In the follicle cells, by contrast, clusters produce piRNAs from one strand, and transposon fragments are strongly biased in the anti-sense direction relative to transcription [62]. Fly ovaries thus produce piRNAs targeting transposons by two distinct mechanisms.

In the *Drosophila* germline, the dominant piRNA clusters are bound by the HP1 homologue Rhino (figure 3), which forms a complex with the linker Deadlock [63–67]. Deadlock recruits Moonshiner and TRF2 (TATA box binding protein related factor 2), which promotes RNA polymerase II (RNA Pol II) transcription from both strands of these transposon-rich loci [68]. Rhino co-localizes with the DXO homologue Cuff, which functions with Rhino to suppress splicing, poly-adenylation and premature termination of piRNA precursor transcripts [64,66,69]. This block to processing may help differentiate piRNA precursors from gene transcripts, as unspliced cluster transcripts are bound by the DEAD box protein UAP56 and the THO complex, which are required for piRNA biogenesis. The resulting piRNA precursor complexes may deliver cluster transcripts to nuclear pores for export to the cytoplasm for processing [70–72].

Most of the piRNA processing machinery, along with the piRNA binding PIWI proteins Aub and Ago3, localizes to perinuclear nuage granules [62]. However, the endonuclease Zuc and a partner protein Papi localize to the mitochondrial outer membranes, and the helicase Armi localizes to both nuage and mitochondria [73]. Precursors are cleaved by Ago3 localized to nuage, or by the mitochondrial nuclease Zuc, which generate intermediates that are the substrates for phased piRNA production [74,75]. This process generates intermediates with defined 5′-ends, and 3′-end extensions that are trimmed by the Nibbler exonuclease, or cleaved by Ago3

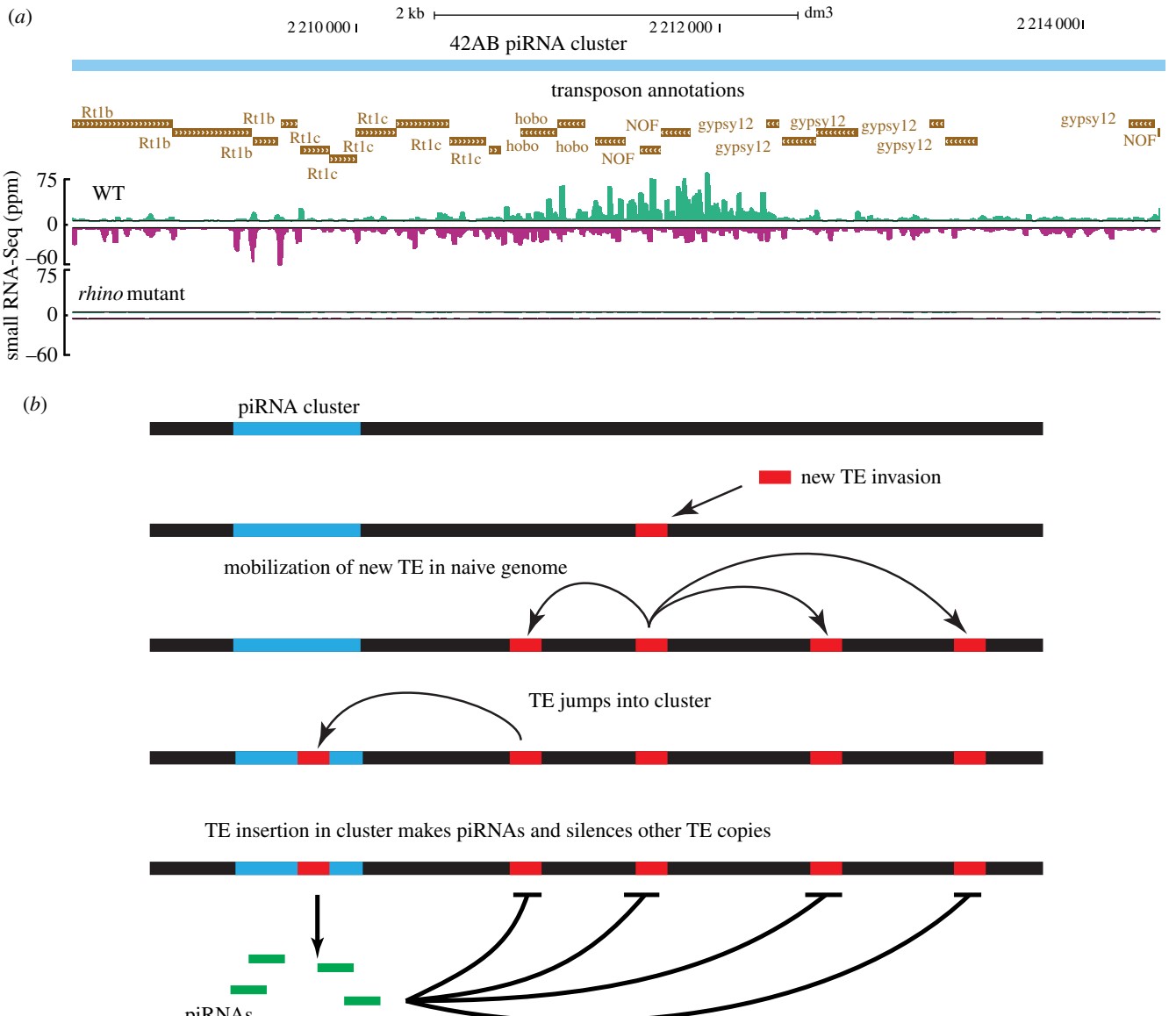

**Figure 2.** Organization of piRNA clusters and possible conflicts with transposons. (a) Genome browser view of 42AB piRNA cluster in D. melanogaster showing transposon organization within the cluster and piRNA levels in wild type and rhino mutant ovaries. (b) Cluster conflicts with transposons. When a transposon jumps into a new species, it can transpose fast and make multiple copies. Once the new transposon jumps into a cluster, piRNAs are generated against the new transposon and these piRNAs can silence the new TE. Thus, it is advantageous for clusters to incorporate new TEs but is deleterious for the transposons.

[76]. The resulting full-length piRNAs are 2′-O-methylated by Hen1, which enhances stability [77].

Somatic follicle cells that surround the developing oocyte also express piRNAs that are required for transposon silencing, but somatic piRNA biogenesis is independent of the RDC (Rhino, Deadlock and Cutoff) complex [64], are derived from clusters that are transcribed on only one strand, and produce capped, spliced and poly-adenylated transcripts that appear similar to mRNAs [64,78,79]. As noted above, the transposon fragments in these 'uni-strand' clusters are strongly biased toward the antisense direction, while the transposon fragments in germline dual-strand clusters are randomly oriented.

Somatic piRNA cluster transcripts are processed in cytoplasmic complexes called Yb bodies, which are distinct from nuage granules: they do not contain Ago3, Aub or Vasa, which are germline specific, but accumulate Yb, Armi, Zuc and Vret. Yb is specific to the somatic pathway, while Armi, Zuc and Vret function in both germline and

somatic piRNA biogenesis [80–84]. piRNA intermediates from the uni-strand flamenco cluster localize next to these Yb bodies [83], and Piwi protein lacking a nuclear localization signal accumulates to the Yb bodies [85]. In addition, the Yb body proteins are required for piRNA loading into Piwi [84]. Yb bodies thus appear to be sites for Piwi–piRNA complex assembly. Within this complex, Zuc cleaves precursors, generating long intermediates with 5′-ends that appear to be bound by Piwi and correspond to the 5′-ends of mature piRNAs [74,75]. The 3′-ends of piRNAs are produced by Zuc-mediated cleavage and trimming to final length by Nibbler [76]. The resulting, Piwi-bound piRNAs are 2′-O-methylated by Hen1 [77]. The 5′-end of the product of initial Zuc cleavage can then be bound by Piwi, and another round of cleavage and trimming produces another piRNA. This process is repeated, generating phased primary piRNAs. These phased piRNAs, bound to Piwi, enter the nucleus and transcriptionally silence TEs [86].

**Figure 3.** piRNA biogenesis mechanisms in flies, mice and worms. Simplified models for piRNA biogenesis are shown for *D. melanogaster* (flies), *M. musculus* (mice) and *C. elegans* (worms). (*a*) Transcription at piRNA clusters, (*b*) the piRNA biogenesis process and (*c*) transposon silencing by piRNAs. In the fly germline, piRNA clusters are marked by H3K9me3. The transcription by RNA Pol II is facilitated by RDC complex along with Moonshiner and TRF2. The transcripts are bound by TREX and UAP56. In the cytoplasm, these transcripts are processed into piRNAs by a ping-pong amplification cycle. The phased piRNAs are further bound by Piwi which can lead to transcriptional silencing by directing histone modification. In the soma, piRNA clusters are transcribed by canonical promoter by RNA Pol II. These transcripts are processed into piRNAs in the Yb bodies present in the cytoplasm and loaded into Piwi, which enters the nucleus to silence transposons. The Slicer cleavage by PIWI Argonautes is shown by vertical scissors and Zucchini mediated cleavage leading to phased piRNAs is shown by tilted scissors. In mice, A-MYB acts as a transcription factor for pachytene piRNA cluster transcription by RNA Pol II. No such transcription factor is known for pre-pachytene piRNA clusters. The ping-pong cycle in the pre-pachytene stage leads to an amplification of piRNAs. piRNA bound MIWI2 can silence transposons by directing DNA methylation. In the pachytene stage, MIWI Slicer cleavage leads to post-transcriptional silencing of transposons. In worms, each piRNA cluster encodes for one piRNA and has its own promoter identified by the Ruby motif. The RNA Pol II mediated transcription is directed by Forkhead and TOFU proteins. Recognition of target by piRNA bound PRG-1 leads to a generation of 22G-RNAs. These secondary 22G-RNAs can mediate transcriptional silencing when in complex with worm specific Argonautes (WAGOs).

## 2.1.2. Secondary piRNA biogenesis

In the germline, piRNAs are amplified by a ping-pong biogenesis cycle [61,87]. Aub binds to piRNAs derived from clusters, which are antisense to active transposons. These complexes cleave transposon transcripts and produce the precursors for sense strand piRNAs bound to Ago3. Ago3 bound to sense strand piRNAs then cleaves antisense piRNA cluster transcripts, producing the precursors of piRNAs that are loaded into Aub, thus completing the cycle. This secondary ping-pong cycle is regulated by the DEAD box helicase Vasa and Tudor domain protein Qin [88,89]. Precursor cleavage by Ago3–piRNA complexes can also produce the substrates of phased piRNAs production. So the ping-pong amplification by Aub and Ago3 in the cytoplasm feeds into the production of phased piRNAs that are bound by Piwi [74,75].

Aub and Ago3 are active endonucleases and cleave targets to post-transcriptionally silence transposons [61,87]. Piwi, by contrast, localizes to the nucleus, where it directs transcriptional silencing through Panoramix and Asterix, which direct repressive histone modification of Piwi targets

[59,90,91]. This is proposed to involve co-transcriptional recognition of nascent transposon transcripts by Piwi–piRNA complexes, but this has not been experimentally confirmed.

## 2.1.3. Primary and secondary biogenesis make the piRNA system adaptive

piRNA clusters appear to function as transposon sequence archives [60,61], and the structure of these domains suggests a simple model for production of primary piRNAs targeting an invading element (figure 2*b*). When a new transposon invades the germline, there are no matching copies in clusters or corresponding piRNAs, and the element is active. Transposition compromises genome integrity and fertility and continues until a copy inserts into a piRNA cluster. The sequence is then incorporated into cluster transcripts, producing piRNAs that trans-silence full-length elements that are dispersed throughout the genome. Subsequent invasion by the same element, or a close relative, would presumably lead to rapid silencing. Furthermore, the secondary amplification system appears to be designed to respond to the

increased expression of an existing element. For example, if a resident element acquires a promoter mutation that increases transcription, the resulting transcripts will feed into the ping-pong cycle, increasing antisense piRNAs and suppressing expression. The *Drosophila* germline piRNA pathway is therefore poised to respond to the new genome invaders, and to changes in the resident genome pathogens.

## 2.2. piRNA biogenesis in mouse

The piRNA pathways in mice and flies share a number of features but also show striking differences [37,92]. In both systems, mutations that disrupt the pathway lead to transposon over-expression and sterility. In *Drosophila*, however, most piRNA pathway mutants are female sterile but male fertile [63,93], with mutations in a few genes leading to both male and female sterility [48,94]. In mice, by contrast, the well-characterized piRNA mutations are male sterile and female fertile [46,47], although recent studies indicate that transposon over-expression in *maelstrom* mutants leads to fetal oocyte attrition [95]. Mice, like flies, have three PIWI clade Argonaute proteins, MILI, MIWI2 and MIWI. piRNAs are also produced from clusters by a Dicer-independent mechanism, and mature piRNAs carry 2′-O-Me modifications at their 3′-ends [46,47,96–99]. Mice also produce two classes of piRNAs. However, they are both expressed in the germline, but at different stages of spermatogenesis (pre-pachytene and pachytene) (figure 3).

During the pre-pachytene stages, mouse piRNAs are enriched in transposon sequences, derived from both genomic strands, and are biased toward a 10 nt overlap, which is characteristic of ping-pong amplification [47]. Amplification of pre-pachytene piRNAs appears to be initiated by MILI bound to primary piRNAs, which cleave target transcripts and generate the precursors of secondary piRNAs bound by MIWI2, which direct DNA methylation and silencing of transposon repeats. This process re-establishes methylation patterns that are erased during the initial developmental stages [37,92]. Flies do not have the DNA methylation machinery, but the ping-pong cycle appears to drive piRNA loading into Piwi, which enters the nucleus and directs repressive H3K9me3 modification of target transposons [36]. The mouse pre-pachytene and fly germline piRNA pathways thus employ similar biogenesis and silencing strategies. However, most transposon mapping piRNAs in the mouse appear to be derived from dispersed elements, not transposon-rich clusters [47], and critical components of the fly biogenesis machinery are not conserved in mouse, including *rhino, deadlock, cuff* and *panoramix*, which function in piRNA precursor production and transposon silencing.

The transition to pachytene in mouse coincides with a significant change in piRNA expression and sequence composition. Expression of the A-MYB transcription factor coordinately activates expression of the Miwi and several hundred piRNA clusters, which produce extremely abundant piRNAs [47,100]. The pachytene clusters are expressed on one genomic strand as either single transcription units (uni-directional clusters) or divergent transcription units (bi-directional clusters). The cluster transcripts are capped, spliced and poly-adenylated, in striking contrast to the germline piRNAs in flies [37,100]. Unlike mouse pre-pachytene and *Drosophila* piRNAs, mouse pachytene piRNA sequences are overwhelmingly unique, and pachytene clusters are de-enriched for transposons and other repeats. A number of studies suggest that these piRNAs regulate protein coding genes, presumably through imperfect base pairing [101,102]. However, pachytene piRNAs accumulate well before global protein coding genes are downregulated late in spermatogenesis, and a direct role for these piRNAs in controlling gene expression has not been conclusively established.

## 2.3. piRNA biogenesis in *C. elegans*

The *C. elegans* piRNA pathway has some similarities and a number of notable differences from other organisms. As observed in other systems, piRNA pathway mutations in *C. elegans* compromise fertility and lead to transposon over-expression [103,104]. However, sterility is not observed in first generation mutant animals, but progressively develops over multiple generations [105,106]. As noted above, piRNAs in flies and mice range from 23 to 30 nt and are derived from long precursor transcripts. In *C. elegans,* by contrast, the piRNAs bound to the PIWI protein, PRG-1, are uniformly 21 nt long and begin with a U [103,104,107,108]. These '21 U RNAs' are produced in a Dicer-independent fashion and carry a 2′-O-methyl modification produced by the Hen1 homologue HENN-1 [109–111], but each *C. elegans* piRNA is produced from a single monocistronic gene, with its own promoter identified by the Ruby sequence motif [108]. Most of these piRNA genes are clustered on the fourth chromosome [108,112–114]. The piRNA genes are transcribed by RNA Pol II, which is regulated by Forkhead and TOFU (Twenty-One-u Fouled Ups) proteins [113,115]. *C. elegans* piRNAs are not amplified by a ping-pong cycle. Instead, PRG-1/21U-RNAs bind to transcripts and prime production of secondary 22G-RNA precursors by RNA-dependent RNA polymerase (RdRP). These secondary piRNAs are bound by worm-specific Argonaute proteins (WAGOs) [116]. These can further mediate transcriptional silencing via repressive histone modification H3K9me3 [38,117,118].

## 3. Diversity in piRNA biogenesis mechanisms

The piRNA pathway clearly has a conserved function of transposon silencing, but diverse mechanisms have evolved to achieve this function. This is reflected in the striking lack of conservation for genes that are often absolutely essential in one system. In *D. melanogaster*, Rhino and Deadlock promote piRNA cluster transcription and transcript processing [64], but homologues are not found in distant species. Mice use A-MYB to regulate transcription of pachytene piRNA clusters, which produce long precursor transcripts [100]. Worms, by contrast, use Forkhead and TOFU to regulate transcription of single piRNA genes, which are not related to piRNA clusters in other organisms [119]. In flies and mice, piRNAs are amplified by a ping-pong cycle [36,92], whereas *C. elegans* piRNA amplification is achieved by RdRP [116]. What drives diversification of the piRNA molecular machinery, and conservation of the core biological function?

The combination of rapid molecular evolution and conservation of core function is typical of a 'Red Queen' host−pathogen arms race. For example, a pathogen gene encoding the target of a host inhibitor mutates to evade

silencing, leading to pathogen replication. Compromised host fitness then drives the selection of mutations in the host gene that restore the interaction and pathogen control. This leads to rapid evolution of the interaction surface, and cycles in which either the pathogen (host is sick) or host (pathogen is silenced) is 'winning' the race. The result is rapid evolution of the interacting proteins, which retain their original functions. In *Drosophila*, a significant subset of piRNA pathway genes is evolving rapidly, under positive selection (reviewed in [6]). Obbard *et al*. [120] calculated ratios of non-synonymous ($K_A$) to synonymous ($K_S$) substitutions for all genes between the sibling species *D. melanogaster* and *D. simulans*, and found that the piRNA pathway genes *krimper, maelstrom, aubergine, piwi, armitage* and *spnE* showed elevated $K_A/K_S$ values, and Lee & Langley [121] found evidence for adaptive evolution of *rhino, krimper, maelstrom, aubergine, armitage, vasa* and *spindle-E*. Simkin *et al*. [122] used phylogenetic analysis by maximum likelihood (PAML) to analyse the evolution of 10 piRNA pathway genes in six *Drosophila* species, and observed positive selection among *rhino, aubergine* and *krimper* genes across multiple *Drosophila* lineages. Signatures of adaptive evolution are also found for piRNA pathway genes in teleost fishes [123], suggesting that the rapid evolution of piRNA genes is widespread.

These findings suggest that the piRNA pathway is engaged in an arms race with transposons, but a typical Red Queen arms race leads to rapid coevolution of host genes encoding proteins that directly interact with the pathogen, and the pathogen genes encoding the targets of these proteins. The fastest evolving genes in the piRNA pathway, by contrast, function in biogenesis, not target recognition. Furthermore, direct analysis of the functional and biochemical consequences of rapid piRNA gene evolution indicates that adaptation targets interactions between piRNA pathway proteins [65,124]. *D. melanogaster* and *D. simulans* are sibling species that can mate and produce viable but sterile progeny. Rhino and Deadlock are rapidly evolving interacting proteins with essential functions in piRNA production in *D. melanogaster*. Significantly, the *D. simulans rhino* and *deadlock* genes do not rescue the corresponding *D. melanogaster* mutations [65]. In addition, endogenous Rhino and Deadlock co-localize and co-precipitate in both *D. melanogaster* and *D. simulans*, but *D. simulans* Rhino does not co-precipitate with *D. melanogaster* Deadlock. This defect maps to the rapidly evolving shadow domain of Rhino, and X-ray crystal structures of the Rhino−Deadlock interfaces show that compensatory mutations in the two proteins have generated species specific interaction surfaces [124]. Adaptive evolution has therefore targeted a critical interaction between proteins required for piRNA biogenesis, which have no direct interaction with the transposon targets of the pathway.

### 3.1. What drives the rapid evolution of the piRNA biogenesis machinery?

Transposon survival depends on evading piRNA control and replication in the germline, and host fertility requires transposon silencing. This would appear to set the stage for a Red Queen arms race driving coevolution of piRNA pathway and transposon genes, not interacting piRNA biogenesis factors. We speculate that this reflects the unique nature of the 'host−pathogen' recognition by piRNAs.

piRNAs map over the full length of target elements [61]. The extent of silencing appears to be proportional to the number of piRNAs mapping to a transposon, but effective silencing can be achieved with relatively limited coverage [125]. The piRNAs that guide silencing are therefore massively redundant, and a transposon would have to accumulate point mutations over the entire transcription unit (coding and non-coding), without disrupting critical open reading frames, in order to evade silencing. In striking contrast, the active transposons silenced by piRNAs are conserved across species [30,31]. Transposons must therefore employ alternative strategies to evade the piRNA pathway.

### 3.2. Evolution of piRNA clusters

*In Drosophila*, clusters appear to be the source of transposon silencing piRNAs [61], and transposition into a cluster is proposed to trigger the silencing of invading elements. Mutations that promote transposition into clusters would therefore be advantageous for the host, whereas mutations biasing transposition to other genomic regions would favour transposons. Many transposons show target site preferences [126]. For example, P-elements prefer to insert into the promoters of germline expressed genes [127]. Germline piRNA clusters, by contrast, are largely transcribed by a non-canonical mechanism that requires the TRF2 transcription factor [68]. This insertion preference may help P-elements evade piRNA clusters. By contrast, an increase in genomic space dedicated to piRNA clusters would provide the host an advantage, by expanding the target for transposition. Consistent with this possibility, there has been a consistent gain in piRNA clusters during the course of evolution [128,129]. Gain in piRNA clusters also offers an advantage to the host by keeping redundant copies of silencing piRNAs. Consistent with this possibility, the deletion of the promoter for a major pachytene piRNA cluster in mouse does not compromise fertility [130,131]. However, mutations in *flamenco* cluster in flies lead to transposon over-expression and sterility [61,132], and deletion of the mouse *pi6* pachytene cluster promoter reduces brood size [131]. These findings suggest that a subset of piRNA clusters is non-redundant and is essential to fertility, while others are redundant. Alternatively, these clusters may target transposons that have degenerated and are no longer functional due to effective silencing over an evolutionary time scale. However, this class of cluster would provide a memory of former genome invasions and thus lead to resistance to new infection by related transposons. In the absence of a new challenge, however, deletion of these clusters would not produce a phenotype.

### 3.3. Evolution of the biogenesis machinery

Adaptive evolution, reflected in elevated rates of non-synonymous substitution, is widespread among piRNA biogenesis genes [6,120]. As noted above, a typical Red Queen arms race leads to rapid coevolution of host defence proteins and their pathogen targets, but most of the rapidly evolving piRNA pathway genes are involved in biogenesis, not target recognition, and adaptive evolution can directly alter protein−protein interactions in the biogenesis pathway [65,124]. These findings suggest that the piRNA pathway may be targeted through molecular mimicry, which has been observed

royalsocietypublishing.org/journal/rsob Open Biol. 9: 180181

royalsocietypublishing.org/journal/rsob    Open Biol. **9**: 180181

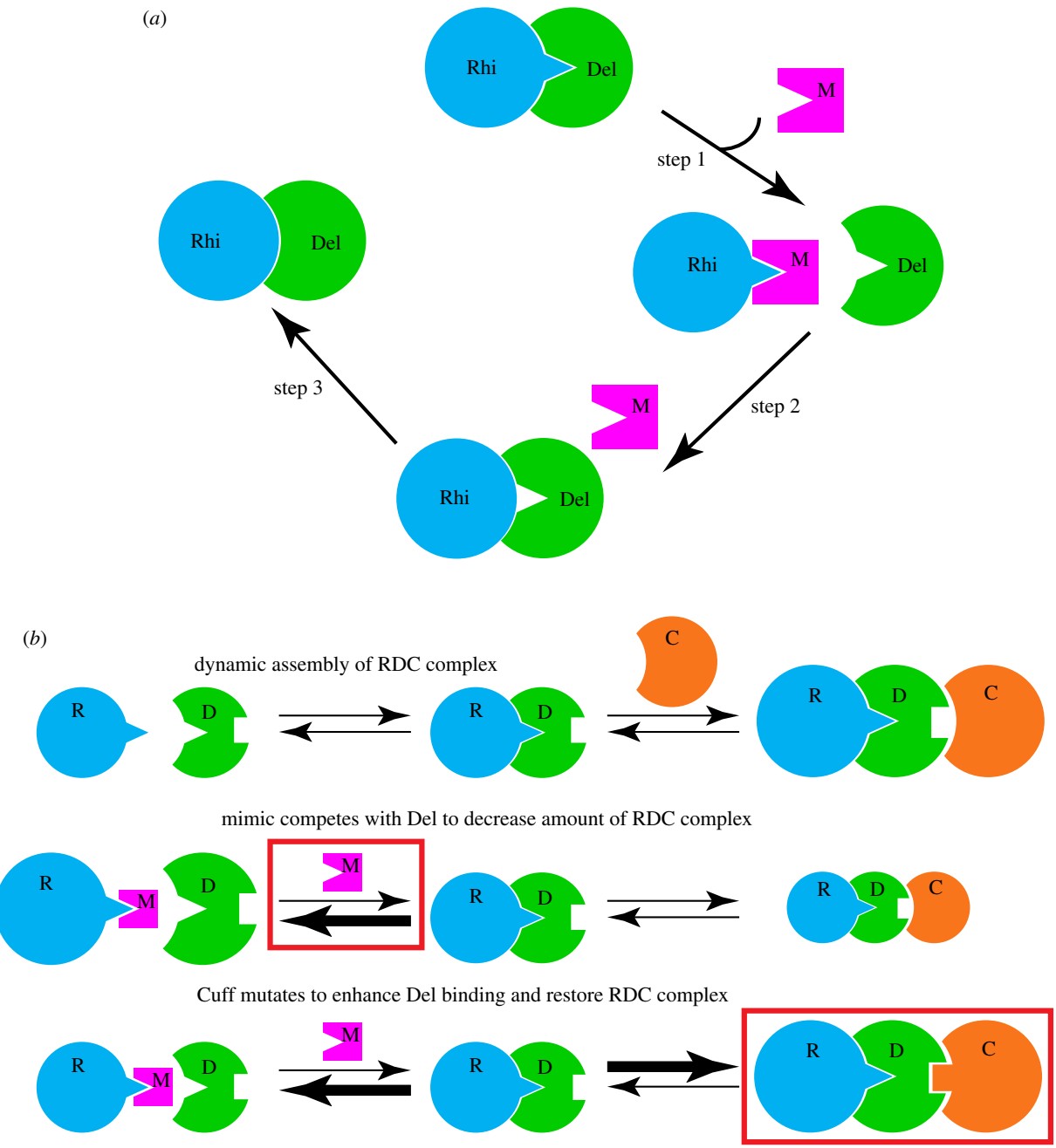

**Figure 4.** A model for piRNA pathway evolution by transposon-encoded mimics. (*a*) In *Drosophila*, Rhino (Rhi) and Deadlock (Del) interact and bind to piRNA clusters, promoting transcription and piRNA processing, and the Rhi–Del interface is rapidly evolving. We speculate that the evolution of this interface was driven by a transposon-encoded protein that mimics the Rhi-binding surface of Del. Step 1. The mimic (M) competes for Rhi, inhibiting biogenesis. Step 2. Selection acts on Rhi mutations that reduce mimic binding, increasing Rhi–Del interactions, but at a cost of reduced affinity. Step 3. Del mutations then restore full Rhi binding. (*b*) Speculative model for coupled evolution of the RDC complex. Rhi (R) binds to Del (D), and the Rhi–Del complex recruits Cutoff (Cuff, C) to form the RDC. A mimic that competes for productive Rhi–Del binding would shift the equilibrium to the left, reducing RDC levels, compromising piRNA production. RDC levels could be restored by Rhi mutations, as in (*a*), or by Cuff mutations that enhance binding to the Rhi–Del complex. In this model, a mimic targeting a single interaction within a coupled system could lead to a cascade of compensatory mutations, altering binding between other proteins in the complex.

in multiple host–pathogen systems [2,133]. In this process, a pathogen protein 'mimics' a host protein surface that interacts with a binding partner in the defence system. The mimic thus competes for the productive interaction, allowing pathogen propagation. This leads to the selection of host mutations that evade mimic binding, often at a cost of reduced binding to the wild-type partner. Selection can then act on mutations at the interaction surface that restore full binding. Mimics thus drive the evolution of protein–protein interactions within the host defence system. We therefore speculate that a protein expressed by an invading transposon, or by a mutant protein produced by a resident element, mimicked the Deadlock surface that interacts with Rhino, competing for productive Rhino–Deadlock binding, and triggered increased transposition. This would lead to the selection of *rhino* mutants that reduce mimic binding, sacrificing affinity for Deadlock. Compensatory mutations in *deadlock* then restored wild-type binding (figure 4*a*). Supporting this view, crystallographic analysis of the Rhino–Deadlock interface in *D. melanogaster* and *D. simulans* reveals a series of compensatory substitutions that generate species-specific binding [124].

The same strategy may target post-transcriptional and transcriptional silencing by piRNAs. Aubergine is rapidly evolving and the *D. simulans* protein shows compromised

royalsocietypublishing.org/journal/rsob    Open Biol. 9: 180181

function in *D. melanogaster* [134]. This PIWI protein binds antisense piRNAs and directs post-transcriptional transposon silencing. Aubergine-mediated cleavage of transposon transcripts is also required for the amplification of the piRNA pool by the ping-pong cycle. Significantly, the DEAD box protein Vasa and the PIWI protein Argonaute3 function with Aubergine during ping-pong amplification, and both of these proteins also show signatures of adaptive evolution [6,122]. Aubergine, Arognaute3 and Vasa also co-localize in perinuclear nuage granules [88], raising the possibility that these cytoplasmic components of the pathway, like Rhino and Deadlock, are co-evolving.

A number of additional genes in the *Drosophila* piRNA pathway show signatures of adaptive evolution [6], and it seems unlikely that transposons encode the variety of mimics needed to directly drive the evolution of all of these targets. However, these rapidly evolving proteins function within higher order nuclear and cytoplasmic assemblies [36], and we speculate that a mimic targeting one interaction within an assembly could be bypassed by mutations in a number of different linked interaction surfaces, provided they restore the level of critical complexes. For example, *Drosophila* Rhino binds to chromatin through H3K9me3, and to Deadlock. Deadlock interacts with Cuff, forming a chromatin bound complex (the RDC, figure 4*b*, top) which appears to be critical to piRNA biogenesis. Within this coupled system, a mimic that competes for Rhino binding to Deadlock would drive the equilibrium toward free Rhino and Deadlock, and reduce the level of the RDC assembly (figure 4*b*, middle). This could be countered by a mutation in Rhino that reduces mimic binding (figure 4*a*). However, a Cuff mutation that increases affinity for the Rhino–Deadlock complex would also drive the equilibrium toward RDC assembly, suppressing the biogenesis defect (figure 4*b*, bottom). In this model, a transposon mimic that targets one interaction in a coupled equilibrium could drive 'coupled evolution' of multiple proteins within the same biochemical pathway. Direct biochemical analysis of additional rapidly evolving piRNA pathway genes will be needed to test this hypothesis.

Alternatively, Blumenstiel *et al*. [6] proposed that competition between effective target silencing and 'autoimmunity' may explain the adaptive evolution of the piRNA pathway. In this model, the piRNA pathway has to retain the ability to adapt to a continuously changing burden of TEs, which are rapidly spread by horizontal and vertical transmission (sensitivity), without targeting protein coding genes, leading to an 'autoimmune response'. This could lead to the selection of mutations that increase the length of piRNAs, increasing the specificity of post-transcriptional and transcriptional silencing. They also proposed that adaptive evolution of the RDC complex may be driven by the need to localize these proteins to piRNA clusters and not at genes, which is required to maintain transposon silencing and to prevent autoimmunity.

# 4. Is reproductive isolation linked to piRNA pathway adaptation?

Reproductive isolation allows speciation, and multiple mechanisms have been proposed to play a role in this process [135]. We speculate that two distinct forms of piRNA pathway adaptation to transposon invasion also contribute to

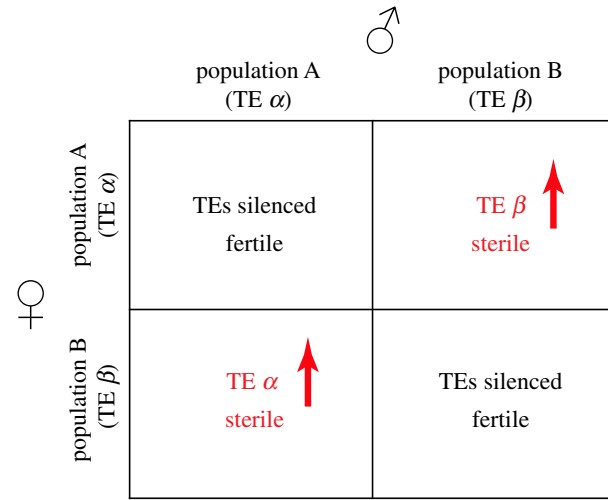

**Figure 5.** Transposon variation can drive reproductive isolation. Hypothetical scenario for crosses between populations A and B having unique transposons. Population A has transposon $\alpha$, but not $\beta$ and population B has transposon $\beta$, but not $\alpha$. Crosses between populations lead to an activation of transposons coming from the father due to the absence of maternal piRNAs against TE unique to father and sterility. This would establish reproductive barriers and lead to a speciation event.

reproductive isolation. The acute response to transposon invasion in *Drosophila* suggests one mechanism. In this system, transposition into a piRNA cluster triggers sequence incorporation into cluster transcript and piRNA production [136], and the piRNAs that silence transposons in the early embryo are maternally deposited [67,137]. As a result, crossing males that carry a 'new' transposon to naive females triggers hybrid dysgenesis, a sterility syndrome caused by activation of the male-specific transposon, as well as resident transposons [136,138]. By contrast, crosses between females that have adapted to a new element and naive males produce fertile offspring, as the new invader is silenced by maternal piRNAs. Adaptation to a single element thus leads to a directional reproductive barrier. However, consider the following scenario: population A has adapted to transposon $\alpha$, but not transposon $\beta$, and population B has adapted to transposon $\beta$, but not $\alpha$ (figure 5). Crosses in either direction between these populations would produce sterile F1 progeny, due to activation of $\alpha$ or $\beta$ transposon, producing a reproductive barrier between animals with identical protein coding genes, but differing in transposon content.

Longer term adaptive evolution of piRNA pathway genes could also drive reproductive isolation. As reviewed above, Rhino–Deadlock coevolution has produced species specific interfaces that prevent functional interactions between proteins from the sibling species *D. melanogaster* and *D. simulans* [65]. Intriguingly, hybrids between these species are sterile, and phenocopy piRNA pathway mutations [134,139]. Together, these observations suggest that biochemical incompatibility between piRNA pathway proteins, driven by adaptive evolution, may directly contribute to reproductive isolation/hybrid sterility. As outlined above, piRNA biogenesis shows remarkable phylogenetic diversity. This could reflect a direct link between the adaptive evolution of the piRNA pathway and reproductive isolation. Emergence of a transposon-encoded mimic would trigger a global burst of transposon mobilization, involving all active elements, due to compromised production of all piRNA

sequences. Intriguingly, bursts of transposition are linked to species divergence in animals and plants [140,141]. An arms race between the piRNA pathway and transposons could therefore generate biochemical incompatibilities that set up reproductive barriers. In addition, this conflict could simultaneously generate transposition induced genetic diversity that can be acted on by natural selection.

## 5. Concluding remarks

Transposons are genomic pathogens which cause genomic instability, and piRNAs have a conserved role in protecting the genomes from transposons. However, the piRNA machinery is rapidly evolving, and many components are poorly conserved. This may result from a Red Queen arms race between transposons and the piRNA pathway, which contributes to genome evolution and generates reproductive barriers and genetic diversity that drive speciation.

Data accessibility. This article has no additional data.

Competing interests. We declare that we have no competing interests.

Funding. This work was supported by National Institute of Child Health and Human Development (R01HD049116 and P01HD078253).

Acknowledgements. We would like to thank the members of the Theurkauf lab, UMass Medical School RNA biology community and Harmit Malik for their insightful discussions.

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
