## [Reviewer comments · Open Biology]

Review History

RSOB-18-0181.R0 (Original submission)

Review form: Reviewer 1

Recommendation

Accept with minor revision (please list in comments)

Are each of the following suitable for general readers?

- a) **Title**
Yes
- b) **Summary**
Yes
- c) **Introduction**
Yes

Is the length of the paper justified?

Yes

Should the paper be seen by a specialist statistical reviewer?

No

Is it clear how to make all supporting data available?

Not Applicable

Is the supplementary material necessary; and if so is it adequate and clear?

Not Applicable

Do you have any ethical concerns with this paper?

No

Comments to the Author

A very Nice review. I suggest the authors refine their manuscript with the following suggestions. Most important is to refine on what actual molecular mimicry is in terms of biochemistry, which the authors are not using here in the actual way that biochemist have defined it, and to also discuss just how essential are piRNA clusters and how quickly the clusters themselves are evolving as a component of the Piwi pathway.

Page 5 and throughout the text, Please fix the naming schemes: PIWI (caps) is the protein or the domain name, piwi (italics) is the gene, and "Piwi pathway" or "Piwi proteins" is the general class name for the pathway and family of proteins.

In abstract, "that's" should be spelled out "that is"

Page 3. Some, not all, retrotransposons show clear relation to retroviruses, especially those with LTRs. But are LINEs also clearly derived from retroviruses since they lack LTRs or clear gag,pol,env genes? Authors should look into this and clarify this paragraph.

Section on transposons as a pathogen, this is a bit of a stretch to apply this term to P-elements and hybrid dysgenesis, there is not an active transmission from one infected host to another target host. Suggest better clarification of this. I would instead suggest focusing the "pathogen" concept of transposons on the ZAM particles that follicle cells and transmit to the *Drosophila* oocyte, and the recent studies of transmissible transposon retroviruses amongst clams mollusks.

Check proper grammar, like "These initial findings" and fixing the awkward language in the paragraph before section 3.1.1.

On sections 3.1.3 and the role of the piRNA clusters, there should also be a discussion on the debate on requirement and rapid evolution of the major clusters themselves, since they seem to be very young amongst *Drosophilids*, not more than 10MYA. There is not yet published data but described in meetings flies are fine with whole clusters deleted, preprints that major piRNA cluster deletion mutants in mice do not show phenotype, and in flies, just a de novo inserted TE by itself can eventually generate its own piRNAs.

Figure 2. piRNA biogenesis mechanisms in different model organisms.

Rest of legend is much too long, where this much commentary belongs in the main text.

Figure 4. Model for piRNA pathway evolution by molecular mimicry. Is this the correct use of

this term molecular mimicry? I think this is usually reserved for when one very disparate molecule type 'mimic's another molecule type, like how translation termination factor proteins or the puromycin antibiotic mimics the charged tRNA. This diagram and concept is more appropriately described as parasite evolving a countermeasure to evade host silencing, like how a virus evolves its epitope or inhibitors that block B-cell recognition.

Review form: Reviewer 2

Recommendation

Accept with minor revision (please list in comments)

Are each of the following suitable for general readers?

- a) **Title**
Yes
- b) **Summary**
Yes
- c) **Introduction**
Yes

Is the length of the paper justified?

Yes

Should the paper be seen by a specialist statistical reviewer?

No

Is it clear how to make all supporting data available?

Not Applicable

Is the supplementary material necessary; and if so is it adequate and clear?

Not Applicable

Do you have any ethical concerns with this paper?

No

Comments to the Author

Parhad and Theurkauf provide a brief review of the piRNA system in fruit flies, *C. elegans*, and mice, and discuss the rapid evolution of some of the piRNA regulatory genes. The piRNA system has been widely reviewed, but the combination of discussing three different organisms along with the evolutionary patterns is distinct. The exploration of molecular mimicry to follow up the Rhi/Deadlock studies is also new.

Overall the review is clear although there are quite a few small grammatical and missing word typos (particularly in figure legends), and some repetition (including between the text and Fig. 2 legend).

Ref. 6 should be cited for discussing the potential role for TE-derived antagonists in driving piRNA system positive selection, although it did conclude that such effects are unlikely to be

pervasive. It would also be useful to briefly mention the autoimmunity hypothesis of those authors as an alternative (not mutually exclusive).

Figure 5 and associated text is essentially a model where TE-induced hybrid dysgenesis leads to sterility between species. In Ref. 129 we aimed to test whether hybrid dysgenesis occurs between species, and concluded that TE activation in hybrids does not correlate with maternal vs. paternal piRNA loading, which is believed to be a major factor in dysgenesis. There's certainly room for more investigation of this issue in other species, but would encourage to mention.

The authors detail some of the major differences in the piRNA system between the 3 animal systems, yet offer few thoughts on how or why these have evolved. It seems unlikely to be driven by the kind of Red Queen dynamics that the authors focus on. Is it differences in TE loads, germline biology differences, other differences in genome organization?

p.2 last paragraph – Has a confusing structure since Helitrons are also DNA transposons.

p.3/4 – Implies that horizontal transfer requires virus-like particles, but then lists P as an example, which doesn't form particles.

p.4 – identified 23-30 nt

p.5 – “dominant clusters” means “predominant clusters”? “random transposon arrays” means “arrays with random orientations”?

p.6 – localizes to perinuclear nuage

p.7 – thus appears to be sites

Signed,
Daniel Barbash
Cornell University

Decision letter (RSOB-18-0181.R0)

14-Nov-2018

Dear Professor Theurkauf

We are pleased to inform you that your manuscript RSOB-18-0181 entitled "Rapid evolution and conserved function of the piRNA genome immune system" has been accepted by the Editor for publication in Open Biology. The reviewer(s) have recommended publication, but also suggest some minor revisions to your manuscript. Therefore, we invite you to respond to the reviewer(s)' comments and revise your manuscript.

Please submit the revised version of your manuscript within 14 days. If you do not think you will be able to meet this date please let us know immediately and we can extend this deadline for you.

When submitting your revised manuscript, you will be able to respond to the comments made by

the referee(s) and upload a file "Response to Referees" in "Section 6 - File Upload". You can use this to document any changes you make to the original manuscript. In order to expedite the processing of the revised manuscript, please be as specific as possible in your response to the referee(s).

- 1) A text file of the manuscript (doc, txt, rtf or tex), including the references, tables (including captions) and figure captions. Please remove any tracked changes from the text before submission. PDF files are not an accepted format for the "Main Document".
- 2) A separate electronic file of each figure (tiff, EPS or print-quality PDF preferred). The format should be produced directly from original creation package, or original software format. Please note that PowerPoint files are not accepted.
- 3) Electronic supplementary material: this should be contained in a separate file from the main text and meet our ESM criteria (see <http://royalsocietypublishing.org/instructions-authors#question5>). All supplementary materials accompanying an accepted article will be treated as in their final form. They will be published alongside the paper on the journal website and posted on the online figshare repository. Files on figshare will be made available approximately one week before the accompanying article so that the supplementary material can be attributed a unique DOI.

Online supplementary material will also carry the title and description provided during submission, so please ensure these are accurate and informative. Note that the Royal Society will not edit or typeset supplementary material and it will be hosted as provided. Please ensure that the supplementary material includes the paper details (authors, title, journal name, article DOI). Your article DOI will be 10.1098/rsob.2016[*last 4 digits of e.g. 10.1098/rsob.20160049*].

- 4) A media summary: a short non-technical summary (up to 100 words) of the key findings/importance of your manuscript. Please try to write in simple English, avoid jargon, explain the importance of the topic, outline the main implications and describe why this topic is newsworthy.

Images

Data-Sharing

It is a condition of publication that data supporting your paper are made available. Data should be made available either in the electronic supplementary material or through an appropriate repository. Details of how to access data should be included in your paper. Please see <http://royalsocietypublishing.org/site/authors/policy.xhtml#question6> for more details.

Data accessibility section

- DNA sequences: Genbank accessions F234391-F234402
- Phylogenetic data: TreeBASE accession number S9123
- Final DNA sequence assembly uploaded as online supplemental material

- Climate data and MaxEnt input files: Dryad doi:10.5521/dryad.12311

Sincerely,

The Open Biology Team
mailto:openbiology@royalsociety.org

Reviewer(s)' Comments to Author:

Referee: 1

Comments to the Author(s)

A very Nice review. I suggest the authors refine their manuscript with the following suggestions. Most important is to refine on what actual molecular mimicry is in terms of biochemistry, which the authors are not using here in the actual way that biochemist have defined it, and to also discuss just how essential are piRNA clusters and how quickly the clusters themselves are evolving as a component of the Piwi pathway.

Page 5 and throughout the text, Please fix the naming schemes: PIWI (caps) is the protein or the domain name, piwi (italics) is the gene, and "Piwi pathway" or "Piwi proteins" is the general class name for the pathway and family of proteins.

In abstract, "that's" should be spelled out "that is"

Page 3. Some, not all, retrotransposons show clear relation to retroviruses, especially those with LTRs. But are LINEs also clearly derived from retroviruses since they lack LTRs or clear gag,pol,env genes? Authors should look into this and clarify this paragraph.

Section on transposons as a pathogen, this is a bit of a stretch to apply this term to P-elements and hybrid dysgenesis, there is not an active transmission from one infected host to another target host. Suggest better clarification of this. I would instead suggest focusing the "pathogen" concept of transposons on the ZAM particles that follicle cells and transmit to the *Drosophila* oocyte, and the recent studies of transmissible transposon retroviruses amongst clams mollusks.

Check proper grammar, like "These initial findings" and fixing the awkward language in the paragraph before section 3.1.1.

On sections 3.1.3 and the role of the piRNA clusters, there should also be a discussion on the debate on requirement and rapid evolution of the major clusters themselves, since they seem to be very young amongst *Drosophilids*, not more than 10MYA. There is not yet published data but described in meetings flies are fine with whole clusters deleted, preprints that major piRNA cluster deletion mutants in mice do not show phenotype, and in flies, just a de novo inserted TE by itself can eventually generate its own piRNAs.

Figure 2. piRNA biogenesis mechanisms in different model organisms.

Rest of legend is much too long, where this much commentary belongs in the main text.

Figure 4. Model for piRNA pathway evolution by molecular mimicry. Is this the correct use of this term molecular mimicry? I think this is usually reserved for when one very disparate molecule type 'mimic's another molecule type, like how translation termination factor proteins or the puromycin antibiotic mimics the charged tRNA. This diagram and concept is more appropriately described as parasite evolving a countermeasure to evade host silencing, like how a virus evolves its epitope or inhibitors that block B-cell recognition.

Referee: 2

Comments to the Author(s)

Parhad and Theurkauf provide a brief review of the piRNA system in fruit flies, *C elegans*, and mice, and discuss the rapid evolution of some of the piRNA regulatory genes. The piRNA system has been widely reviewed, but the combination of discussing three different organisms along with the evolutionary patterns is distinct. The exploration of molecular mimicry to follow up the Rhi/Deadlock studies is also new.

Overall the review is clear although there are quite a few small grammatical and missing word typos (particularly in figure legends), and some repetition (including between the text and Fig. 2 legend).

Ref. 6 should be cited for discussing the potential role for TE-derived antagonists in driving piRNA system positive selection, although it did conclude that such effects are unlikely to be pervasive. It would also be useful to briefly mention the autoimmunity hypothesis of those authors as an alternative (not mutually exclusive).

Figure 5 and associated text is essentially a model where TE-induced hybrid dysgenesis leads to sterility between species. In Ref. 129 we aimed to test whether hybrid dysgenesis occurs between species, and concluded that TE activation in hybrids does not correlate with maternal vs. paternal piRNA loading, which is believed to be a major factor in dysgenesis. There's certainly room for more investigation of this issue in other species, but would encourage to mention.

The authors detail some of the major differences in the piRNA system between the 3 animal systems, yet offer few thoughts on how or why these have evolved. It seems unlikely to be driven by the kind of Red Queen dynamics that the authors focus on. Is it differences in TE loads, germline biology differences, other differences in genome organization?

p.2 last paragraph - Has a confusing structure since Helitrons are also DNA transposons.

p.3/4 - Implies that horizontal transfer requires virus-like particles, but then lists P as an example, which doesn't form particles.

p.4 - identified 23-30 nt

p.5 - "dominant clusters" means "predominant clusters"? "random transposon arrays" means "arrays with random orientations"?

p.6 - localizes to perinuclear nuage

p.7 - thus appears to be sites

Signed,

Daniel Barbash

Cornell University

Author's Response to Decision Letter for (RSOB-18-0181.R0)

See Appendix A.

Decision letter (RSOB-18-0181.R1)

03-Dec-2018

Dear Dr Theurkauf

We are pleased to inform you that your manuscript entitled "Rapid evolution and conserved function of the piRNA genome immune system" has been accepted by the Editor for publication in Open Biology.

Sincerely,

The Open Biology Team
mailto:openbiology@royalsociety.org

Appendix A

Comments to reviews:

We would like to thank esteemed reviewers for their suggestions to improve the manuscript. We have highlighted our response in **bold** below:

Referee: 1

Comments to the Author(s)

A very Nice review. I suggest the authors refine their manuscript with the following suggestions. Most important is to refine on what actual molecular mimicry is in terms of biochemistry, which the authors are not using here in the actual way that biochemist have defined it, and to also discuss just how essential are piRNA clusters and how quickly the clusters themselves are evolving as a component of the Piwi pathway.

These suggestions are addressed below in their respective sections.

Page 5 and throughout the text, Please fix the naming schemes: PIWI (caps) is the protein or the domain name, piwi (italics) is the gene, and ³Piwi pathway² or ³Piwi proteins² is the general class name for the pathway and family of proteins.

We have fixed the naming scheme.

In abstract, ³that¹s² should be spelled out ³that is²

Fixed.

Page 3. Some, not all, retrotransposons show clear relation to retroviruses, especially those with LTRs. But are LINEs also clearly derived from retroviruses since they lack LTRs or clear gag, pol, env genes? Authors should look into this and clarify this paragraph.

The origin of LINE elements is not known. To make the association of LTR transposon and retrovirus clear, we have moved the part

describing this association to the section on LTR transposons. (page 3, paragraph 1)

Section on transposons as a pathogen, this is a bit of a stretch to apply this term to P-elements and hybrid dysgenesis, there is not an active transmission from one infected host to another target host. Suggest better clarification of this. I would instead suggest focusing the ³pathogen² concept of transposons on the ZAM particles that follicle cells and transmit to the Drosophila oocyte, and the recent studies of transmissible transposon retroviruses amongst clams mollusks.

We have included above references in the revised manuscript. ZAM ref#21 (page 3, paragraph 1), clam ref#26 (page 3, paragraph 2)

Check proper grammar, like ³These initial findings² and fixing the awkward language in the paragraph before section 3.1.1.

Fixed.

On sections 3.1.3 and the role of the piRNA clusters, there should also be a discussion on the debate on requirement and rapid evolution of the major clusters themselves, since they seem to be very young amongst Drosophilids, not more than 10MYA. There is not yet published data but described in meetings flies are fine with whole clusters deleted, preprints that major piRNA cluster deletion mutants in mice do not show phenotype, and in flies, just a de novo inserted TE by itself can eventually generate its own piRNAs.

As this topic involves describing both fly and mouse piRNA clusters, we have described it in depth in section 4.2 (Evolution of piRNA clusters) after introducing clusters in both the organisms. We have included the cluster deletion discussion as suggested by the reviewer. (Page 12, paragraph 2)

Figure 2. piRNA biogenesis mechanisms in different model organisms. Rest of legend is much too long, where this much commentary belongs in the main text.

As this single figure compares the piRNA pathway in three different model organisms, and we have been unable to significantly shorten it without loss of clarity.

Figure 4. Model for piRNA pathway evolution by molecular mimicry. Is this the correct use of this term molecular mimicry? I think this is usually reserved for when one very disparate molecule type mimics another molecule type, like how translation termination factor proteins or the puromycin antibiotic mimics the charged tRNA. This diagram and concept is more appropriately described as parasite evolving a countermeasure to evade host silencing, like how a virus evolves its epitope or inhibitors that block B-cell recognition.

We propose that a transposon encoded protein (gag, pol or env) mimics the region of Del that binds to Rhino. While all of the molecules involved are proteins, the proteins involved have very disparate functions. We believe this is a form of molecular mimicry, and explains how a transposon encoded protein could drive evolution of a protein-protein interaction with the piRNA biogenesis pathway. We have rewritten the figure legend to clarify this point.

Referee: 2

Comments to the Author(s)

Parhad and Theurkauf provide a brief review of the piRNA system in fruit flies, *C. elegans*, and mice, and discuss the rapid evolution of some of the piRNA regulatory genes. The piRNA system has been widely reviewed, but the combination of discussing three different organisms along with the evolutionary patterns is distinct. The exploration of molecular mimicry to follow up the Rhi/Deadlock studies is also new.

Thank you!

Overall the review is clear although there are quite a few small grammatical and missing word typos (particularly in figure legends), and some repetition (including between the text and Fig. 2 legend).

As this single figure shows the comparison of piRNA pathway in different model organisms, it contains a lot of information and we have tried to explain at least the diagrammed piRNA components in the figure legend. It would be difficult to follow this figure with a few details in the figure legend.

Ref. 6 should be cited for discussing the potential role for TE-derived antagonists in driving piRNA system positive selection, although it did conclude that such effects are unlikely to be pervasive. It would also be useful to briefly mention the autoimmunity hypothesis of those authors as an alternative (not mutually exclusive).

We now described this hypothesis on page 14, paragraph 2.

Figure 5 and associated text is essentially a model where TE-induced hybrid dysgenesis leads to sterility between species. In Ref. 129 we aimed to test whether hybrid dysgenesis occurs between species, and concluded that TE activation in hybrids does not correlate with maternal vs. paternal piRNA loading, which is believed to be a major factor in dysgenesis. There's certainly room for more investigation of this issue in other species, but would encourage to mention.

Similar to interspecific hybrids, resident transposon activation is also observed in case of P-M dysgenics (Ref. 136). We think that the activation of one or a few transposons could affect the entire piRNA pathway machinery and lead to activation of many more TEs than just predicted by maternal piRNAs. To highlight the complexity of the speciation process, we have made it clear that a piRNA-transposon arms race could be one of the many mechanisms leading to reproductive isolation (page 14, paragraph 3).

The authors detail some of the major differences in the piRNA system between the 3 animal systems, yet offer few thoughts on how or why these have evolved. It seems unlikely to be driven by the kind of Red Queen dynamics that the authors focus on. Is it differences in TE loads, germline biology differences, other differences in genome organization?

We think that the continuous nature of the transposon-piRNA arms race, leading to essentially endless cycles of piRNA pathway evolution, could have produced the diverse piRNA biogenesis mechanisms currently operating. We have added this description on page 11, paragraph 1. We have been unable to devise other mechanism, but welcome suggestions.

p.2 last paragraph - Has a confusing structure since Helitrons are also DNA transposons.

We have now explicitly mentioned that helitrons are also DNA transposons.

p.3/4 - Implies that horizontal transfer requires virus-like particles, but then lists P as an example, which doesn't form particles.

This is an excellent point that we should have addressed. We now clarify the P elements do not form virus like particles, and that the mechanism of horizontal transfer is not understood.

p.4 - identified 23-30 nt

Fixed.

p.5 - ³dominant clusters² means ³predominant clusters²? ³random transposon arrays² means ³arrays with random orientations²?

Fixed.

p.6 - localizes to perinuclear nuage

Fixed.

p.7 - thus appears to be sites

Fixed.

Signed,

Daniel Barbash

Cornell University